# Characterization of the Subclinical Infection of Porcine Deltacoronavirus in Grower Pigs under Experimental Conditions

**DOI:** 10.3390/v14102144

**Published:** 2022-09-28

**Authors:** Lu Yen, Juan Carlos Mora-Díaz, Rolf Rauh, William Nelson, Gino Castillo, Fangshu Ye, Jianqiang Zhang, David Baum, Jeffrey Zimmerman, Rahul Nelli, Luis Giménez-Lirola

**Affiliations:** 1Department of Veterinary Diagnostic and Production Animal Medicine, College of Veterinary Medicine, Iowa State University, Ames, IA 50011, USA; 2Tetracore, Inc., Rockville, MD 20850, USA; 3Department of Statistics, College of Liberal Arts and Sciences, Iowa State University, Ames, IA 50011, USA

**Keywords:** coronavirus, deltacoronavirus, viral shedding, immune response, subclinical infection, grower pigs

## Abstract

This study characterized the susceptibility and dynamic of porcine deltacoronavirus infection in grower pigs under experimental conditions using a combination of syndromic and laboratory assessments. Seven-week-old conventional pigs (*n* = 24) were randomly distributed into PDCoV- (*n* = 12) and mock-inoculated (*n* = 12) groups. Serum was collected at −7, 0, 3, 7, 10, 14, 17, 21, 28, 35, and 42 days post-inoculation (DPI) to evaluate viremia (RT-qPCR) and antibody response (S1-based ELISA). Viral shedding and potential infectivity were determined using pen-based oral fluids and feces collected every other day between DPI 0 and 42. Pigs showed no clinical signs or viremia throughout the study. Active virus shedding was detected in feces (6-22 DPI) and oral fluids (2-30 DPI), peaking at DPI 10. IgG was first detected at DPI 10, being statistically significant after DPI 14 and increasing thereafter, coinciding with the progressive resolution of the infection. Likewise, a significant increase in proinflammatory IL-12 was detected between DPI 10 and 21 in PDCoV-inoculated pigs, which could enhance innate resistance to PDCoV infection. This study demonstrated that active surveillance based on systematic sampling and laboratory testing combining molecular and serological tools is critical for the accurate detection of subclinical circulation of PDCoV in pigs after weaning.

## 1. Introduction

Porcine deltacoronavirus (PDCoV) is a member of the subfamily *Coronavirinae* of the genus *Deltacoronavirus* [1] that was initially identified in Hong Kong in 2009 during a molecular surveillance study [2]. PDCoV is phylogenetically related to avian deltacoronaviruses detected in domestic and wild birds [2,3,4], and it is the only known deltacoronavirus capable of infecting and causing disease in pigs.

PDCoV is considered a transboundary and emerging pathogen. Since 2014, this virus has been consistently associated with diarrheal outbreaks in different countries in North and South America [5,6,7,8], mainland China [9,10,11] and other parts in Asia [12,13,14].

Like the other two major enteropathogenic porcine coronaviruses, transmissible gastroenteritis virus (TGEV) [15] and porcine epidemic diarrhea virus (PEDV) [16], PDCoV replicates in small intestinal enterocytes, causing enteric disease in suckling pigs, characterized by profuse watery diarrhea, emesis, and subsequent dehydration [17]. Likewise, although less severe, the gross and histologic changes in the gut of pigs infected with PDCoV are similar to those observed in TGEV [18] and PEDV [19]. Lesions are characterized by thin translucent walls distended with yellow fluid, normally confined to the small intestine causing atrophic enteritis and subsequent reduction or failure in the capability to absorb nutrients. Histological observations of the small intestine revealed regions of small villus blunting and fusion and minimal lymphoblastic infiltration of the villi of the lamina propria [17,20,21,22,23,24].

Virus-specific molecular and serological tools may be used to confirm the circulation of PDCoV in swine herds. Through them, PDCoV has been detected in intestinal, fecal, saliva, vomit, environmental, and feed samples via quantitative reverse transcription PCR (RT-qPCR), and confirmed by sequencing [8,25,26]. Moreover, retrospective serological studies suggested that PDCoV could have circulated in pigs before its first identification in the USA [27]. Likewise, PDCoV has been co-detected with other intestinal pathogens, especially PEDV and rotaviruses [25]. The continuous development of and improvement in a broad range of direct and indirect detection methods are critical for differential diagnosis of clinical cases of PDCoV-associated enteric disease [28].

There are no previous reports of PDCoV outbreaks in grower or adult pigs. As with other porcine coronaviruses (PorCoVs), the susceptibility and clinical outcome of the infection seem to be inversely related to the age of the pig, although this has not been proven. In addition, all experimental animal studies carried out to date have been focused on suckling pigs [17,21,23,29]. Thus, the objective of this study was to characterize the susceptibility and dynamics of PDCoV infection in grower pigs under experimental conditions using a combination of syndromic and laboratory methods.

## 2. Materials and Methods

### 2.1. The Study

Seven-week-old grower pigs (*n* = 24) were acquired from a commercial farm that was pre-screened negative for the different PorCoVs circulating in USA swine herds [30]. Pigs were randomly divided into two groups/rooms, 12 pigs per room with 2 pigs per pen (six pens per room). Pigs were inoculated orally with PDCoV (30 mL PDCoV USA/IL/2014 strain, passage 13th, at 1.5 × 10^6^ TCID_50_/mL per pig) or mock-inoculated with culture medium (30 mL per pig, orally). The study was approved by the Institutional Animal Care and Use Committee of Iowa State University (IACUC# 5-15-8017-S, 15 March 2016). Pigs were clinically evaluated twice daily for general health and signs of enteric disease. Serum samples were tested for PDCoV RNA (RT-qPCR; Tetracore Inc., Rockville, MD, USA), antibody (enzyme-linked immunosorbent assay), and cytokine/chemokine (magnetic bead-based multiplex assay) responses to PDCoV infection. Pen feces and oral fluids were tested for the presence of infectious PDCoV and for PDCoV RNA. All pigs were euthanized at DPI 42 using a penetrating captive bolt device (Accles and Shelvoke, Ltd., Sutton Coldfield, UK).

### 2.2. Sample Collection, Processing, and Storage

Pig blood samples were collected at −7, 0, 3, 7, 10, 14, 17, 21, 28, 35, and 42 days post-inoculation (DPI) from the jugular vein or cranial vena cava using a blood collection system (Becton Dickinson, Franklin Lakes, NJ, USA) and vacutainer serum separation tubes (Kendall, Mansfield, MA, USA). Serum was harvested by centrifugation at 1500 × g for 5 min, aliquoted into 2 mL cryogenic vials (Cryovial^®^, Greiner Bio-One, Monroe, NC, USA), and stored at −80 °C until testing.

Pen (2 pigs per pen) feces and oral fluids were collected every other day from DPI 0 to 42. Floor fecal samples were collected in 50 mL conical tubes (Corning^®^, Corning, NY, USA) and aliquoted (~2 mg) in 2 mL cryogenic vials (Greiner Bio-One). Oral fluids were collected as previously described [31,32]. Briefly, 3-strand 1.6 cm 100% cotton rope (Web Rigging Supply, Inc., Carrollton, GA, USA) was hung from a bracket fixed to one side of each pen for 30 min, giving pigs time to chew on and interact with the rope. The rope was then severed, placed in a plastic bag, passed through a clothes wringer (Dyna-Jet, Overland Park, KS, USA), and decanted into 50 mL conical tubes. Samples were aliquoted into 2 mL cryogenic tubes (Cryovial®), Greiner Bio-One) and stored at −80 °C until testing.

### 2.3. Sample Processing for Assessment of virus Infectivity in Cell Culture

Frozen (−80 °C) longitudinal pen-based fecal specimens were thawed, and a sample (~0.5 mg) was taken using applicators (Thermo Fisher Scientific, Waltham, MA, USA) pre-wetted with phosphate-buffered saline (PBS) (Gibco^TM^, Thermo Fisher Scientific) and transferred to snap cap tubes (~0.5 mg) (Thermo Fisher Scientific) containing 0.5 mL of PBS. The snap cap tubes were vortexed to homogenize the feces attached to the applicators with PBS before harvesting the supernatant into a deep 96-well plate (Abgene^TM^, Thermo Fisher Scientific). Likewise, frozen (−80 °C) oral fluids were thawed, vortexed and centrifuged at 2000× *g* for 10 min at 4 °C, followed by the collection of the supernatant with a transfer pipette (VWR, Radnor, PA, USA). The fecal and oral fluids supernatants were transferred into a 96-well 0.20 µm filter plate (Corning) attached to a deep 96-well plate (Abgene^TM^, Thermo Fisher Scientific) and centrifuged at 2000× *g* for 15 min at 4 °C, allowing the supernatant to be filtered to the receptor plate. Filtered supernatants were diluted 10 times with an infection medium to be used as inoculum [33] and kept at 4 °C until being used for cell culture inoculation.

### 2.4. Inoculation of Fecal and Oral Fluid Specimens in Cell Culture

Swine (*Sus scrofa*) testicular cells (ST cells, ATCC CRL-1746, Manassas, VA, USA) were cultured in a 96-well flat clear bottom black polystyrene surface-treated microplate (CellBIND^®^; Corning), with a concentration of 2.5 × 10^4^ cells per well, using growth medium [Advanced Minimum Essential Medium (MEM; Gibco^TM^, Thermo Fisher Scientific) supplemented with 5% of fetal bovine serum (Atlas Biologicals, Fort Collins, CO, USA), 1% HEPES (Gibco^TM^, Thermo Fisher Scientific), 1% GlutaMAX (Gibco^TM^, Thermo Fisher Scientific), 1% penicillin-streptomycin (Gibco^TM^, Thermo Fisher Scientific), and 25 µg/mL of Amphotericin B (Gibco^TM^, Thermo Fisher Scientific)] and incubated at 37 °C with 5% CO_2_. After 24 h (with a confluency > 80%) growth medium was removed, and cells were washed two times with 100 µL of infection medium (Advanced MEM (Gibco^TM^, Thermo Fisher Scientific) supplemented with 1% HEPES (Gibco^TM^, Thermo Fisher Scientific), 1% GlutaMAX (Gibco^TM^, Thermo Fisher Scientific), 1% penicillin-streptomycin (Gibco^TM^, Thermo Fisher Scientific), 25 µg/mL of Amphotericin B (Gibco^TM^, Thermo Fisher Scientific), and 2.5 µg/mL of 0.1% trypsin (Sigma-Aldrich, St. Louis, MO, USA)). Cells were inoculated (in duplicate) with 100 µL of PDCoV (Michigan/8977/2014 strain; National Veterinary Services Laboratories, Ames, IA, USA) at 1.1 × 10^5^ TCID_50_/mL (positive control), infection medium (negative control), and previously frozen fecal or oral fluid specimens, and incubated for 1.5 h at 37 °C with 5% CO_2_ to allow virus adsorption. After two washes with 150 µL of PBS (Gibco^TM^, Thermo Fisher Scientific), and the addition of a new infection medium (100 µL), the cells were incubated at 37 °C with 5% CO_2_. Inoculated cells were observed under an inverted microscope to determine the possible absence (Figure 1A) or presence (Figure 1B) of cytopathic effect (CPE), described as rounded cells, cluster of rounded cells, and cell detachment. After 48 h post-inoculation (hpi), the cells were fixed with 50 µL of 80% acetone for 15 min at room temperature. 

### 2.5. Immunofluorescence Assay

Pen feces and oral fluid samples collected throughout the longitudinal study were evaluated for their potential infectivity (i.e., viable PDCoV) in vitro. Specifically, the expression of PDCoV structural nucleocapsid (N) protein in ST cell monolayers inoculated with fecal and oral fluid specimens collected from PDCoV inoculated pigs was evaluated via IFA. Briefly, the fixed assay plates were washed once with 200 µL of PBS. Then, 100 µL of mouse IgG1 fluorescein isothiocyanate (FITC; green fluorescent)-conjugated anti-PDCoV N protein monoclonal antibody (Medgene, Labs, Brookings, SD, USA) diluted 1:150 in PBS with 0.1% bovine serum albumin (BSA; Jackson Immuno Research, West Grove, PA, USA) were added to each well and the plate was incubated for 1 h at 37 °C with 5% CO_2_. Cells were counterstained using DAPI nuclear staining (blue fluorescent) by adding 100 µL of NucBlue reagent (Invitrogen^TM^, Thermo Fisher Scientific) to each well and incubating for 15 min at room temperature according to manufacturer specifications. After three washes with 200 µL of PBS, the cells were observed under a CKX41 inverted microscope (Olympus Life Science, Waltham, MA, USA), and images were captured with the camera INFINITY 2 (Lumenera, Ottawa, Canada) and the software INFINITY ANALYZE (Lumenera). The absence of green fluorescence (N protein) and sole presence of blue fluorescent intensity (DAPI) was indicative of no virus infection (Figure 1C, E, G). Likewise, the detection of green fluorescent intensity was indicative of virus infection in the ST cells (Figure 1D,F,H). 

### 2.6. Quantitative Reverse Transcription PCR

Reverse-transcription quantitative PCR (RT-qPCR) testing was performed on serum samples (*n* = 264) collected at DPI −7, 0, 3, 7, 10, 14, 17, 21, 28, 35, and 42 for testing viremia in PDCoV- and mock-inoculated pigs. Likewise, the kinetics of the virus shedding (detection of viral RNA) was evaluated on pen feces and oral fluids collected every other day between DPI 0 and 42 from both treatment groups (*n* = 264). Viral RNA extractions were carried out using the MagMAX-96 Pathogen RNA/DNA kit (Thermo Fisher Scientific) with KingFisher Flex 96 Deep-Well Magnetic Particle Processor (Thermo Fisher Scientific) following the manufacturer’s instructions. 

In this study, we used the Tetracore EZ-PED/TGE/PDCoV Real-Time RT-PCR assay (Tetracore, Inc.), designed for simultaneous detection and differentiation of PEDV, TGEV, and PDCoV. Inhibition controls (IC) were added to the lysis buffer to ensure the consistency of the extraction process. Each RT-qPCR reaction (25 μL final volume) consisted of 18 μL of Master Mix ready-to-use (enzyme blend included) and 7 μL of the extracted viral RNA (IC included). Each run included a negative extraction control, individual positive control for PEDV, TGEV, and PDCoV, respectively, and a “no template” control. PCRs were run on an ABI-7500 real-time PCR instrument (Thermo Fisher Scientific) with cycling conditions, 48 °C for 15 min and 95 °C for 2 min holding; 45 cycles, 95 °C for 5 s denaturation, and 60 °C for 40 s amplification. The PCR results were analyzed using ABI 7500 software (v 1.4), and samples with a threshold cycle (Ct) above 40 were considered negative.

### 2.7. Enzyme-Linked Immunosorbent Assay

A PDCoV indirect ELISA based on the amino-terminal subunit (S1) of the glycosylated structural spike protein was used to assess and monitor virus-specific IgG serum response to PDCoV infection in grower pigs. A consensus coding region of the PDCoV S1 gene was synthetically generated, amplified by PCR, cloned in pNPM5 mammalian expression vector and transfected into HEK293 cells for soluble expression, purification by a combination of protein A and nickel affinity chromatography techniques and, after removal of the Fc-tag, further purification and enrichment via molecular exclusion chromatography (GE Healthcare, Pittsburgh, PA, USA), that were fully described in a separate study (unpublished data). Purified PDCoV S1 protein was diluted at 0.6 μg/mL in PBS pH 7.4 and coated onto 96-well plates (Immuno Breakables Modules, Thermo Fisher Scientific, Agawam, MA, USA) and incubated at 4 °C for 16 h. Plates were then washed 5 times with PBS pH 7.4, containing 0.1% Tween 20 (PBST), blocked with a 1% (wt/vol) BSA solution (Jackson ImmunoResearch Inc.), incubated at 25 °C for 2 h, dried at 37 °C for 3 h. Serum samples collected from PDCoV- (*n* = 132) and mock-inoculated control (*n* = 132), as well as positive and negative test controls (in duplicate), were tested at 1:100 (100 μL/well) in goat-serum-based sample diluent (50%), and the plate incubated at 37 °C for 1 h. Then, plates were washed 5 times (350 μL/well) with PBST (PBS pH 7,4 and 0.1% Tween 20), and 100 μL of peroxidase-conjugated goat anti-pig IgG (Fc) at 1:50,000 antibody (Bethyl Laboratories, Inc., Montgomery, TX, USA) were added to each well and incubated at 37 °C for 30 min. The reaction was visualized after 5 min incubation with 100 μL of tetramethylbenzidine-hydrogen peroxide (TMB) substrate solution per well (SurModics IVD, Inc., Eden Prairie, MN, USA) and stopped with 100 μL of stop solution per well (SurModics). Optical density was measured at 450 nm using an ELISA plate reader and SoftMax Pro7 software (Molecular Devices, San Jose, CA, USA). Serum IgG responses were expressed as sample-to-positive (S/P) ratios. In a previous study, we estimated that an S/P cutoff value > 0.25 provides 100% specificity without cross-reactivity with other porcine coronaviruses (unpublished data). However, to increase the sensitivity of the ELISA, samples with S/P values > 0.20 were considered PDCoV IgG antibody positives. 

### 2.8. Multiplex Porcine Cytokine and Chemokine Immunoassay

A porcine cytokine and chemokine 9-plex Luminex^®^ assay, including interferon (IFN)-α, IFN-γ, interleukin (IL)-1β, IL-4, IL-6, IL-8, IL-10, IL-12/IL-23p40, and tumor necrosis factor (TNF)-α (ProcartaPlex™ Panel; Invitrogen, Thermo Scientific, Frederick, MD, USA), was performed as directed by manufacturer’s instructions on serum samples (*n* = 192) collected from PDCoV- and mock-inoculated control pigs at DPI 0, 3, 7, 10, 14, 17, 21, and 28. All reagents were provided by the manufacturer. In brief, 50 µL of ready-to-use capture bead mix was added to each well of a flat-bottom 96-well black plate and washed once with 150 µL of wash buffer. Serum samples and standards were diluted at 1:2 in universal assay buffer and transferred to the test plate. Then, the plate was shaken at 600 revolutions per min (rpm) at room temperature for 30 min, followed by an overnight (16 h) incubation at 4 °C, and an additional incubation at room temperature with shaking for 30 min. After washing the test plate twice with 150 µL of wash buffer, 25 µL of detection antibody solution was added to each well, and the plate was incubated at room temperature for 30 min and shaken at 600 rpm. Subsequently, 50 µL of streptavidin- phycoerythrin was added to the plate after two washes with wash buffer as indicated above and incubated with shaking at room temperature for 30 min. After two washes with wash buffer, the beads were resuspended with 120 µL of reading buffer and shaken at room temperature for 5 min. Testing was performed using a Bio-Plex 200 system operated by the Bio-Plex Manager software (Bio-Rad, Hercules, CA, USA) as follows: 50 µL acquisition volume, 60 s timeout, 5000–25,000 DD gate, and 50 minimum bead count. The fluorescence intensity of each sample was subtracted from the blank wells, and the concentration (pg/mL) of each cytokine/chemokine was calculated from the standard curve generated from the kit’s internal standards and analyzed using GraphPad Prism^®^ 9 (GraphPad Software Inc., La Jolla, CA, USA). 

### 2.9. Data Analysis 

A linear mixed model (PROC Mixed and Slice Statement) was used to analyze the differences in the detection of PDCoV by RT-qPCR (Ct values) between oral fluids and fecal samples collected over time (every other day between 0-42 DPI) from each pen (*n* = 6) of the PDCoV-inoculated group. The analysis was performed using “specimen” (oral fluids and feces) and “DPI” as fixed effects and “pen” as the random effect (each pen was sampled every other day between 0 and 42 DPI). The differences in the percentage of PDCoV RT-qPCR positive pens detected in feces and oral fluids were compared using chi-square test.

A mixed-effect linear model was also used to study the differences in S/P values between PDCoV-inoculated pigs and control pigs for PDCoV-specific IgG responses with “treatment” (PDCoV- vs. mock-inoculated) and “DPI” as fixed effects, and the pig as the random effect. 

Likewise, to analyze the differences in the detection of IFN-α, IL-1β, IL-12, IL-12/IL-23p40, IL-4, IL-6, IL-8, and TNF-α (Luminex) between the PDCoV- and mock-inoculated pigs, a linear mixed model was applied, whereas “group” (Control and PDCoV-inoculated), “DPI”, and their interaction term (Group*DPI) as fixed effects and the pig ID nested within the “group” as a random effect.

Statistical analyses were performed using SAS version 9.4 (SAS Institute, Inc, Cary, NC, USA) and GraphPad Prism^®^ 9 (GraphPad Software Inc.). For all analyses, a *p*-value < 0.05 was considered statistically significant.

## 3. Results

### 3.1. Absence of Noticeable Clinical Signs and Viremia after PDCoV Inoculation but Active Virus Replication and Shedding Detected in Feces and Oral Fluids

As with the mock-inoculated control pigs, neither clinical signs nor viremia was observed or detected in PDCoV-inoculated pigs over the course of the study. Virus shedding (PDCoV RNA) was detected by RT-qPCR in oral fluids from DPI 2 to 20, with a statistically significant (*p* < 0.05) peak in the Ct values observed between DPI 8-10 (Figure 2A). However, no statistically significant (*p* > 0.05) differences in the percentage of PDCoV RT-qPCR positive pens were detected at any time point. PDCoV was also shed in feces between DPI 6 and 22, peaking at DPI 10 (*p* < 0.05) (Figure 2B). As with oral fluids, no significant differences (*p* > 0.05) in the percentage of PDCoV RT-qPCR positive pens were detected. Further analysis comparing shedding levels across specimens showed significantly higher (*p* < 0.05) levels in oral fluid compared to feces at DPI 8 (Appendix A). No viable virus was isolated or detected by IFA on monolayer cultures of ST cells inoculated with previously frozen (−80 °C) fecal and oral fluid samples collected from PDCoV inoculated pigs throughout the study (Table 1).

### 3.2. Specific IgG Seroconversion Detected in Serum Two Weeks after PDCoV Inoculation 

The first serum IgG response to PDCoV was detected by ELISA at 10 DPI in two animals (2/12). With a peak at DPI 21, the PDCoV-specific (S1 protein) IgG response was significantly higher (*p* < 0.05) in the PDCoV group compared to the control group from DPI 14 until the end of the study (DPI 42) (Figure 3). All pigs in the PDCoV-inoculation group seroconverted in response to virus inoculation (i.e., PDCoV IgG positive at any time point throughout the observational period) (Figure 3).

### 3.3. Increased Serum Levels of Proinflammatory Cytokine IL-12 in Response to PDCoV Infection

The results and analysis of the serum levels obtained for the cytokines/chemokines (IFN-α, IFN-γ, IL-1β, IL-4, IL-6, IL-8, IL-10, IL-12, and TNF-α) were assessed using a multiplex Luminex assay are presented in Figure 4. No significant differences in the serum levels of IFN-α (Figure 4A) and IL-1β (Figure 4B) were observed between the PDCoV- and mock-inoculated groups. A significant (*p* < 0.05) increase in IL-12 serum levels was detected 10, 14 and 21 days following PDCoV-inoculation (Figure 4C). Although a slight increase in the IL-4 (Figure 4D), IL-6 (Figure 4E) and TNF-α (Figure 4G) levels were observed at DPI 10 in PDCoV-inoculated pigs, coinciding with the peak of viral shedding, these levels were not statistically significant (*p* > 0.05) compared to those of the mock-inoculated group. With the exception of DPI 3, serum levels of IL-8 were significantly (*p* < 0.05) higher from DPI 0 through DPI 21 (Figure 4F), however, those levels were already significantly higher before virus inoculation on DPI 0. No detectable levels of IFN-γ and IL-10 were observed at any time point and hence not plotted.

## 4. Discussion

Outbreaks of PDCoV enteric disease in neonatal and nursing pigs have been reported in the Americas and Asia [5,6,7,8,9,10,11,12,13,14]. Moreover, different studies were able to replicate PDCoV enteric disease in suckling pigs under experimental conditions [17,20,21,23,29,34,35,36]. The clinical and histopathologic presentation of PDCoV is non-specific, i.e., acute, watery diarrhea due to atrophic enteritis, resembling other enteropathogenic viral infections. Thus, accurate and specific detection of PDCoV infections is key for differential diagnosis from other swine enteric diseases affecting suckling pigs.

The objective of this study was to characterize the susceptibility to and virus dynamics during PDCoV infection in grower pigs under experimental conditions. As with pigs in the mock-inoculated negative control group, PDCoV-inoculated pigs appeared clinically healthy throughout the study. Zhao et al. (2020) [37] reported diarrhea between 4 and 21 DPI in weaned pigs inoculated with PDCoV at 10^4^ TCID_50_/mL, which was lower than the infectious dose (10^6^ TCID_50_/mL) used in our study, but pigs recovered and no obvious gross or histologic lesions were observed at necropsy. Moreover, Vitosh-Sillman et al. (2016) [20] observed transient (5 to 8 DPI) and soft to diarrheic feces in sows exposed to PDCoV via aerosol. However, in line with our study, there are no previous outbreak reports of PDCoV-associated disease in older pigs, and the prevalence and overall impact of PDCoV in older animals remain unknown. Neonatal pigs show a slower turnover of villus epithelium after infection [38], being more susceptible to clinical disease by enteric coronaviruses [39,40].

Only a few PDCoV experimental studies reported transient and short viremia (not more than 5 DPI) in neonatal pigs [20,21,29,35]. Although no viremia was detected in the present study, virus replication and fecal shedding were detected by RT-qPCR between 6 and 22 days following PDCoV inoculation, consistent with an early study on weaned conventional pigs [37]. Previous experimental studies on suckling pigs reported earlier onset of viral shedding (1-3 DPI) in fecal specimens [17,20,29]. Nevertheless, the time from viral shedding onset to peak viral load relative to clinical vs. subclinical presentation and disease severity needs to be further investigated.

Vitosh-Sillman et al. (2016) [20] detected PDCoV shedding in litter-based oral fluids collected between 14 and 28 DPI. However, this study found that the time of detection (DPI 2) to shedding resolution (DPI 30) was longer in pen-based oral fluids than in feces. Likewise, other studies comparing the viral shedding of PEDV [41], TGEV [32], and porcine hemagglutinating encephalomyelitis virus (PHEV) [42] found longer shedding in oral fluids than in feces via RT-qPCR testing. Together, all these studies suggest that oral fluid could be a more appropriate specimen for coronavirus surveillance, particularly in large populations.

In vitro assessment of virus infectivity in ST cells showed no viable virus in any pen-based fecal and oral fluid specimens that tested positive by RT-qPCR throughout the study. This came as no surprise due to the low sensitivity of PDCoV isolation in vitro [28]. Importantly, the specimens collected during the present study showed RT-qPCR Ct values higher than 30 and specimens were kept at −80 °C before testing. Previous reports describing successful isolation of PDCoV highlighted the importance of using fresh specimens with low RT-qPCR Ct values (< 20) and often collected from naïve piglets orally inoculated (“feedback”) with virus-enriched infectious material (intestinal contain or fecal material) [22,33]. 

Besides the virus shedding, PDCoV subclinical infection in grower pigs was further demonstrated by specific seroconversion of PDCoV-inoculated pigs. In line with previous experimental studies [20,29,37], PDCoV IgG response in this study was first detected via indirect ELISA at DPI 10, coinciding with the peak of virus shedding, and increased through DPI 21. The moderate IgG response reported here could be associated with the progressive resolution of the infection.

In this study, serum levels of proinflammatory cytokine IL-12 were significantly increased in the PDCoV-inoculated group between DPI 10 and 21, coinciding with the peak of shedding and IgG response, respectively. Similar findings were reported in gnotobiotic 10-day-old piglets soon after inoculation (DPI 1); however, results from this previous study were based only on one animal [35]. IL-12 is a natural killer (NK) cell stimulatory factor (NSKSF) that induces IFN-γ production via NK cells, Th1 and CD8 cells, and plays an important role in T-cell proliferation [43]. It will be interesting to study the role of NK and T cells during PDCoV infection and across age groups but this is beyond the scope of this study. Unlike the increased levels of IFN-γ between DPI 7 and 35 reported by Zhao et al. (2020) [37] in weaned pigs experimentally inoculated with PDCoV, no detectable serum levels of IFN-γ were observed during the present study. In general, proinflammatory cytokines enhance innate resistance while shaping the ultimate antigen-specific immune response. Interestingly, a previous PEDV infection study showed that pro-inflammatory cytokine profiles in suckling pigs differed from those of weaned pigs and were associated with the onset of fecal PEDV RNA shedding [43]. Unfortunately, there is limited information about the systemic cytokine/chemokine response to PDCoV infection and its association with age, viral shedding, and severity of the disease, which deserve further investigation. 

It is important to be reminded that the absence of clinical disease is not the same as absence of pathogen shedding and the ability to trigger a specific immune response. Under the specific conditions of this study, it was demonstrated that PDCoV inoculation of grower pigs can result in subclinical infection. Asymptomatic carriers can provide a source of a pathogen for other susceptible populations of pigs if fomites are shared among different populations. This can happen on multi-barn sites that are all-in–all-out by barn but not by site. The present study provides new insights on specific tools, and appropriate sampling and testing strategies (trinomial “test–specimen–time”) for accurate detection of PDCoV in pig populations, particularly subclinical circulation in grower and adult animals. 

## Figures and Tables

**Figure 1 viruses-14-02144-f001:**
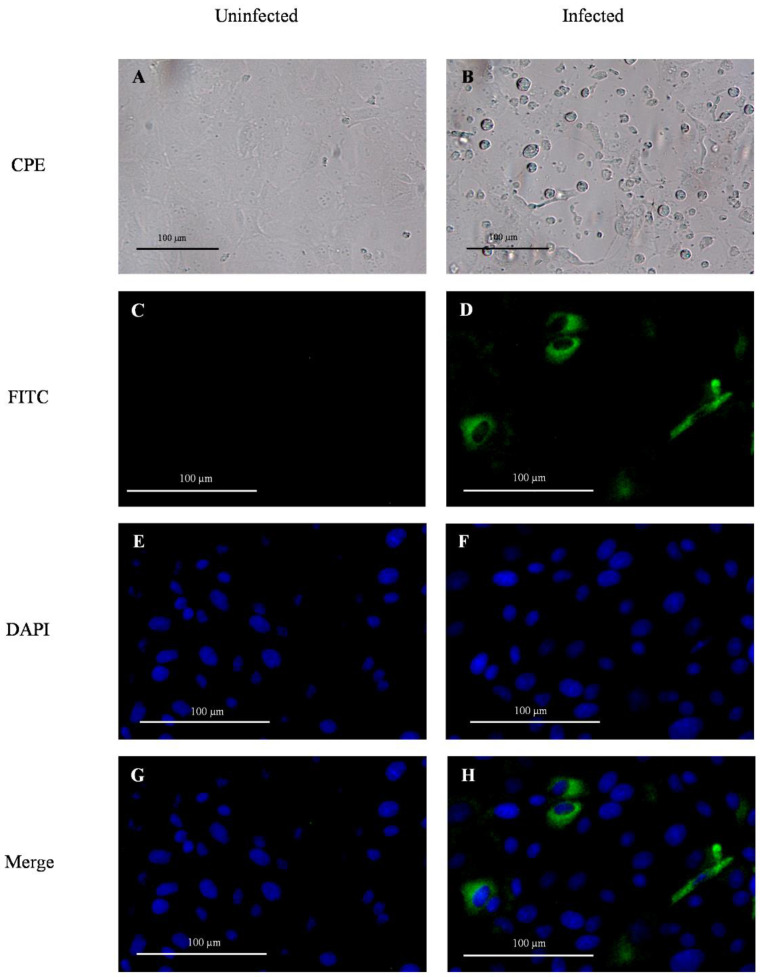
Detection of porcine deltacoronavirus (PDCoV) infection and replication in monolayer cultures of swine testicular (ST) cells in vitro via observation of cytopathic effect under the inverted microscope (CPE), and the expression of viral protein N (green fluorescence) revealed by immunofluorescent antibody assay (IFA). The negative infection control was inoculated with culture medium (**A**,**C**,**E**,**G**), while the positive infection control corresponded to ST cells inoculated with PDCoV (1.1 × 10^5^ TCID_50_/mL) (**B**,**D**,**F**,**H**). PDCoV IFA was performed using mouse IgG1 fluorescein isothiocyanate (FITC; green)-conjugated anti-PDCoV N protein monoclonal antibody (Medgene) diluted 1:150 in phosphate-buffered saline with 0.1% bovine serum albumin and nuclear counterstain with DAPI (blue). Images were obtained using a fluorescence inverted microscope (Olympus Life Science, FCKX41) with the camera INFINITY 2 (Lumenera) at 20X magnification.

**Figure 2 viruses-14-02144-f002:**
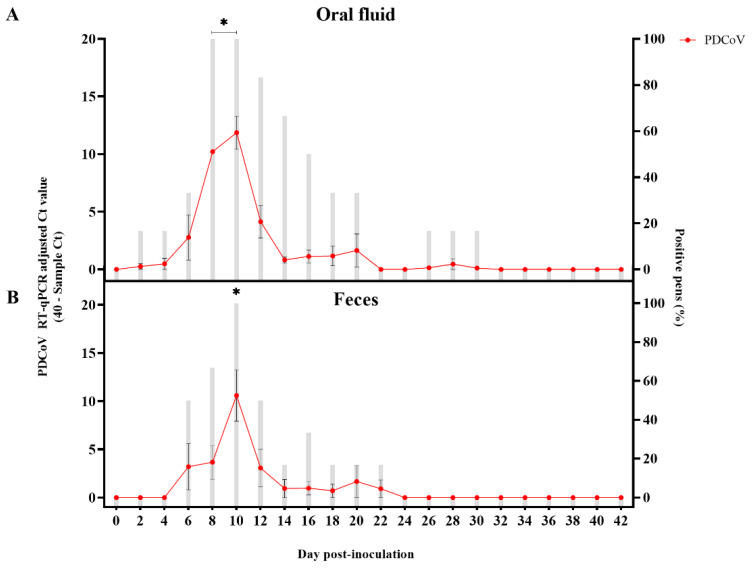
Detection of porcine deltacoronavirus (PDCoV) RNA in pen-based oral fluids (*n* = 6) and fecal (*n* = 6) samples collected every other day between day post-inoculation (DPI) 0 to 42 from PDCoV- and mock-inoculated 7-week-old grower pigs using a commercial porcine enteric coronaviruses multiplex RT-qPCR (EZ-PED/TGE/PDCoV MPX 1.1 (Tetracore Inc.) specific for porcine epidemic diarrhea virus (PEDV), transmissible gastroenteritis virus (TGEV) and PDCoV. This graph represents the average data of 40-sample Ct values on the left *y*-axis (line chart with markers), while the average percentage of positive pens were shown on the right *y*-axis (bar-graph). Samples with Ct values above 40 were considered negative. (**A**) Detection in pen-based oral fluid samples; (**B**) Detection in pen-based fecal samples. Error bars represent the standard error of the mean (SEM), and * denotes statistical significance (*p* < 0.05).

**Figure 3 viruses-14-02144-f003:**
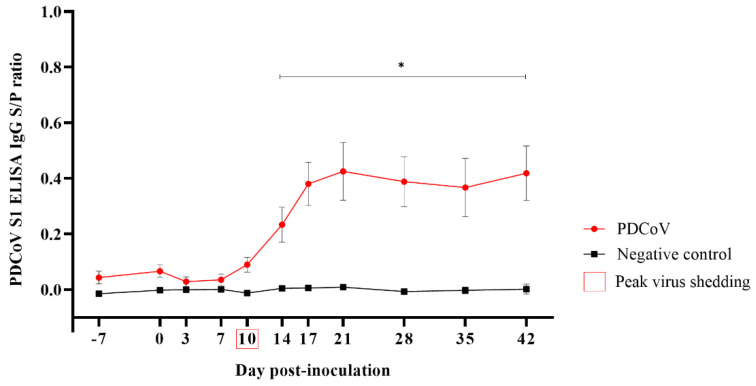
Porcine deltacoronavirus (PDCoV) S1 ELISA IgG responses [mean sample-to-positive (S/P) values, standard error of the mean (SEM)] (*y*-axis) at −7, 0, 3, 7, 10, 14, 17, 21, 28, 35, and 42 days post-inoculation (DPI) (*x*-axis), in 7-week-old grower pigs experimentally inoculated with PDCoV (*n* = 12) or mock-inoculated (*n* = 12) with culture medium. Error bars represent the standard error of the mean SEM, and * denotes statistical differences (*p* < 0.05).

**Figure 4 viruses-14-02144-f004:**
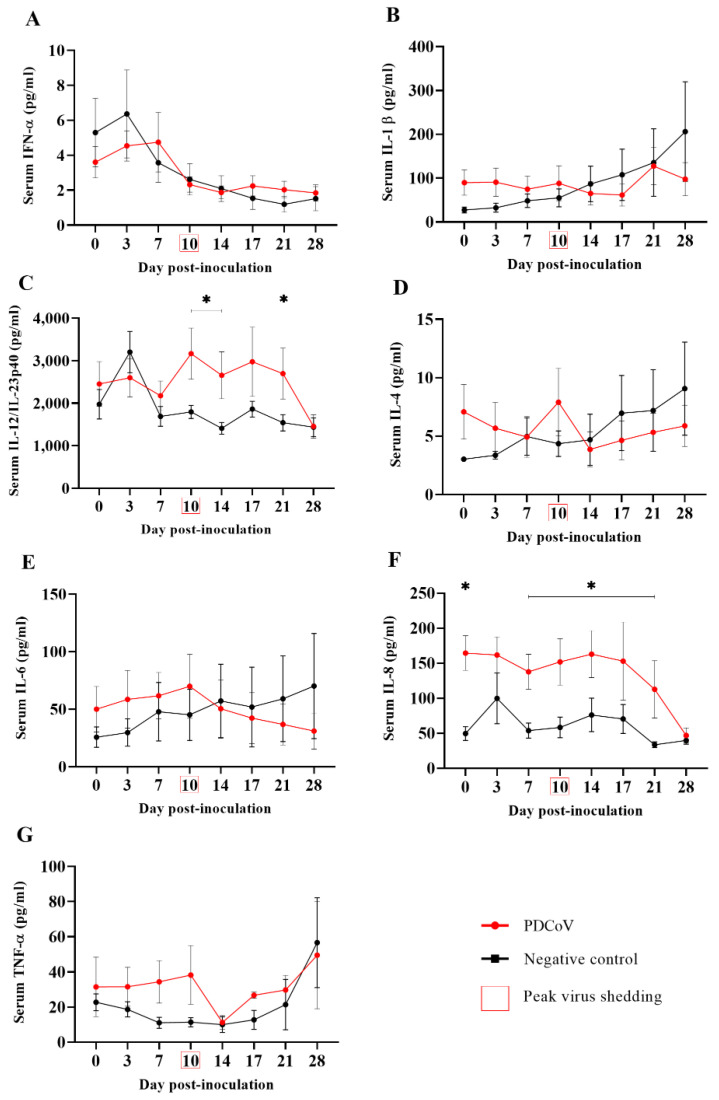
Serum cytokine/chemokine levels (ng/mL; *y*-axis) of IFN-α(**A**), IL-1β (**B**), IL-12/IL-23p40 (**C**), IL-4 (**D**), IL-6 (**E**), IL-8 (**F**), and TNF-α (**G**) over time (*x*-axis) in 7-week-old pigs (*n* = 12) experimentally inoculated with PDCoV, compared to corresponding mock-inoculated pigs (*n* = 12). Serum samples were collected at 0, 3, 7, 10, 14, 17, 21, and 28 days post-inoculation (DPI). Graphs were generated from data collected via multiplex Luminex assay testing (ProcartaPlex™ Panel from Invitrogen). IFN-γ and IL-10 were not detected at any time point and hence were not plotted. The minimum detectable levels for each cytokine/chemokine evaluated were: 0.87 pg/mL (IFN-α), 2.66 pg/mL (IL-1β), 48 pg/mL (IL-12/IL-23p40), 1.56 pg/mL (IL-4), 6.84 pg/mL (IL-6), 4.66 pg/mL (IL-8), 5.86 pg/mL (TNF-α), 4.96 pg/mL (IFNγ), and 4.76 pg/mL (IL-10). Error bars represent the standard error of the mean (SEM), and * denotes statistical differences (*p* < 0.05).

**Table 1 viruses-14-02144-t001:** Detection of porcine deltacoronavirus (PDCoV) RNA (RT-qPCR Ct values) in pen-based oral fluid and feces following PDCoV-inoculation and in vitro assessment of potential specimen infectivity (viable virus) via inoculation of swine testicular (ST) cells.

DPI ^a^	Pen 1			Pen 2			Pen 3			Pen 4			Pen 5			Pen 6		
	Oral Fluid/Feces			Oral Fluid/Feces			Oral Fluid/Feces			Oral Fluid/Feces			Oral Fluid/Feces			Oral Fluid/Feces		
	RT-qPCR ^b^	CPE ^c^	IFA ^d^	RT-qPCR	CPE	IFA	RT-qPCR	CPE	IFA	RT-qPCR	CPE	IFA	RT-qPCR	CPE	IFA	RT-qPCR	CPE	IFA
0	- ^e^/-	-/-	-/-	-/-	-/-	-/-	-/-	-/-	-/-	-/-	-/-	-/-	-/-	-/-	-/-	-/-	-/-	-/-
2	-/-	-/-	-/-	38.5/-	-/-	-/-	-/-	-/-	-/-	-/-	-/-	-/-	-/-	-/-	-/-	-/-	-/-	-/-
4	-/-	-/-	-/-	37.1/-	-/-	-/-	-/-	-/-	-/-	-/-	-/-	-/-	-/-	-/-	-/-	-/-	-/-	-/-
6	35.1/37.4	-/-	-/-	28.2/25.0	-/-	-/-	-/-	-/-	-/-	-/38.4	-/-	-/-	-/-	-/-	-/-	-/-	-/-	-/-
8	29.6/-	-/-	-/-	29.7/37.5	-/-	-/-	29.8/38.7	-/-	-/-	30.0/31.3	-/-	-/-	29.5/30.5	-/-	-/-	30.1/-	-/-	-/-
10	29.7/31.4	-/-	-/-	23.7/20.0	-/-	-/-	31.1/32.1	-/-	-/-	23.6/22.7	-/-	-/-	30.8/35.8	-/-	-/-	30.0/34.4	-/-	-/-
12	36.4/36.9	-/-	-/-	32.8/37.0	-/-	-/-	36.5/-	-/-	-/-	30.9/27.7	-/-	-/-	38.5/-	-/-	-/-	-/-	-/-	-/-
14	38.6/-	-/-	-/-	39.0/-	-/-	-/-	38.9/-	-/-	-/-	38.6/34.3	-/-	-/-	-/-	-/-	-/-	-/-	-/-	-/-
16	38.2/-	-/-	-/-	36.7/38.2	-/-	-/-	38.3/-	-/-	-/-	-/-	-/-	-/-	-/35.9	-/-	-/-	-/-	-/-	-/-
18	-/-	-/-	-/-	34.9/-	-/-	-/-	-/-	-/-	-/-	38.1/35.7	-/-	-/-	-/-	-/-	-/-	-/-	-/-	-/-
20	-/-	-/-	-/-	-/-	-/-	-/-	38.9/-	-/-	-/-	31.2/30.0	-/-	-/-	-/-	-/-	-/-	-/-	-/-	-/-
22	-/-	-/-	-/-	-/-	-/-	-/-	-/-	-/-	-/-	-/34.5	-/-	-/-	-/-	-/-	-/-	-/-	-/-	-/-
24	-/-	-/-	-/-	-/-	-/-	-/-	-/-	-/-	-/-	-/-	-/-	-/-	-/-	-/-	-/-	-/-	-/-	-/-
26	-/-	-/-	-/-	39.1/-	-/-	-/-	-/-	-/-	-/-	-/-	-/-	-/-	-/-	-/-	-/-	-/-	-/-	-/-
28	-/-	-/-	-/-	37.2/-	-/-	-/-	-/-	-/-	-/-	-/-	-/-	-/-	-/-	-/-	-/-	-/-	-/-	-/-
30	-/-	-/-	-/-	39.3/-	-/-	-/-	-/-	-/-	-/-	-/-	-/-	-/-	-/-	-/-	-/-	-/-	-/-	-/-
32	-/-	-/-	-/-	-/-	-/-	-/-	-/-	-/-	-/-	-/-	-/-	-/-	-/-	-/-	-/-	-/-	-/-	-/-
34	-/-	-/-	-/-	-/-	-/-	-/-	-/-	-/-	-/-	-/-	-/-	-/-	-/-	-/-	-/-	-/-	-/-	-/-
36	-/-	-/-	-/-	-/-	-/-	-/-	-/-	-/-	-/-	-/-	-/-	-/-	-/-	-/-	-/-	-/-	-/-	-/-
38	-/-	-/-	-/-	-/-	-/-	-/-	-/-	-/-	-/-	-/-	-/-	-/-	-/-	-/-	-/-	-/-	-/-	-/-
40	-/-	-/-	-/-	-/-	-/-	-/-	-/-	-/-	-/-	-/-	-/-	-/-	-/-	-/-	-/-	-/-	-/-	-/-
42	-/-	-/-	-/-	-/-	-/-	-/-	-/-	-/-	-/-	-/-	-/-	-/-	-/-	-/-	-/-	-/-	-/-	-/-

^a^ DPI, days post-inoculation; ^b^ RT-qPCR, Quantitative reverse transcription PCR; ^c^ CPE, Cytopathic effect; ^d^ IFA, Immunofluorescence assay; ^e^ -, non-detected.

## Data Availability

All data are available from the corresponding author upon reasonable request.

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
