# Peer review of "Characterization of the Subclinical Infection of Porcine Deltacoronavirus in Grower Pigs under Experimental Conditions"

_viruses, 2022, doi:10.3390/v14102144_

Round 1
Reviewer 1 Report
The manuscript by Yen et al present the subclinical infection by porcine deltacoronavirus in growing pigs (age: 7 week old). Virus is replicating and shed orally and fecal. The lack of symptoms is an intriguing finding. It is of interest to the scientific community. The paper is clearly written. One minor suggestion for improvement is provided below
- Figure 2: legend in the figure indicates that TGEV and PEDV loads were also determined, and I assume that these are all negative. However, the samples being all negative is very difficult (impossible) to judge from the figure, and if indeed all negative this may also be only mentioned in the text. Now, the grey bars of the right Y-axe can mistakenly be seen as signals of the TGEV PCR (also grey). Suggestion is to delete the data of TGEV and PEDV from the figure.
Author Response
Dear Reviewer
Thank you for your review. We agree with your suggestion, therefore we have changed Fig. 2 as recommended.
Sincerely,
Luis G. Gimenez-Lirola
Associate Professor
Iowa State University
Reviewer 2 Report
In this manuscript, the authors describe the susceptibility and dynamics of PDCoV infection in grower pigs under experimental conditions, combining symptoms observation and laboratory methods. This information is of great reference value, so it is recommended to be published in Viruses. As for the analysis of cytokines, it is recommended to use SPF pigs to reduce the impact from other microorganisms.
Author Response
Dear Reviewer
Thank you for your review. Although we agree on that the cytokine response could be somehow impacted by other microorganisms, it was very important to perform this study on conventional pigs to mimic field conditions.
Thank you again for your time
Sincerely,
Luis G. Gimenez-Lirola
Associate Professor
Iowa State University